# A cross-sectional study on the impact of the COVID-19 pandemic on psychological outcomes: Multiple indicators and multiple causes modeling

**Musheer A. Aljaberi**[1,2,3]*, **Naser A. Alareqe**[4], **Abdulsamad Alsalahi**[5], **Mousa A. Qasem**[6], **Sarah Noman**[2], **Md. Uzir Hossain Uzir**[7], **Lubna Ali Mohammed**[8], **Zine.El. Abiddine Fares**[9], **Chung-Ying Lin**[10], **Atiyeh M. Abdallah**[11], **Rukman Awang Hamat**[12]*, **Mohd Dzulkhairi Mohd Rani**[13]*

**1** Faculty of Medicine and Health Sciences, Taiz University, Taiz, Yemen, **2** Department of Community Health, Faculty of Medicine and Health Sciences, Universiti Putra Malaysia, Serdang, Malaysia, **3** Faculty of Nursing and Applied Sciences, Lincoln University College, Petaling Jaya, Malaysia, **4** Department of Educational Psychology, Faculty of Education, Taiz University, Taiz, Yemen, **5** Department of Pharmacology, Faculty of Pharmacy, Sana'a University, Sana'a, Yemen, **6** Department of Pharmaceutical Technology, Faculty of Pharmacy, University of Malaya, Kuala Lumpur, Malaysia, **7** Faculty of Business Economics and Social Development, Universiti Malaysia Terengganu, Kuala Terengganu, Terengganu, Malaysia, **8** Faculty of Social Science, Arts, and Humanities, Lincoln University College, Petaling Jaya, Malaysia, **9** Psychological and Educational Research Lab, Department of Psychology, University of Djillali Liabes, Sidi Bel Abbes, Algeria, **10** Institute of Allied Health Sciences, College of Medicine, National Cheng Kung University, Tainan, Taiwan, **11** Department of Biomedical Sciences, College of Health Sciences, QU-Health, Qatar University, Doha, Qatar, **12** Department of Medical Microbiology, Faculty of Medicine and Health Sciences, Universiti Putra Malaysia, Serdang, Malaysia, **13** Department of Primary Health Care, Faculty of Medicine and Health Sciences, Universiti Sains Islam Malaysia, Nilai, Malaysia

* drdzulkhairi@usim.edu.my (MDMR); rukman@upm.edu.my (RAH); musheer@upm.edu.my (MAA)

**Data Availability Statement:** Regarding the availability of the datasets that support this study's

## Abstract

Although the psychological impact of coronavirus disease 2019 (COVID-19) has been evaluated in the literature, further research is needed, particularly on post-traumatic stress disorder (PTSD) and psychological outcomes, is needed. This study aims to investigate the effect of the COVID-19 pandemic on psychological outcomes (depression, anxiety, and insomnia). A cross-sectional study using an online survey was conducted using the following instruments: Impact of Event Scale-Revised (IES-R), Patient Health Questionnaire-9 (PHQ-9), Generalized Anxiety Disorder (GAD-7), and Insomnia Severity Index (ISI). Confirmatory factor analysis (CFA), structural equation model (SEM), multiple indicators and multiple causes (MIMIC) modeling, and differential item functioning (DIF) were performed to analyze the collected data. According to the results, participants with PTSD (n = 360) showed a higher level of depression, anxiety, and insomnia than those without PTSD (n = 639). Among the participants, 36.5% experienced moderate to severe symptoms of depression, and 32.6% had mild depressive symptoms. Moreover, 23.7% of participants experienced moderate to severe anxiety symptoms, and 33.1% had mild anxiety symptoms. In addition, 51.5% of participants experienced symptoms of insomnia. In conclusion, the PTSD caused by COVID-19 is significantly associated with depression, anxiety, and insomnia at the level of latent constructs and observed variables.

findings, we indicated in the consent form for the participants that their response is confidential and will protect throughout the data collection and analysis to guarantee data integrity and privacy. We have also stated that the data is for exclusive scientific use, only for research and academic purposes upon request. Therefore, for interested researchers to apply to gain access to the data, the data can be accessed upon reasonable request for academic and research use from the following Institution details as follows: Institution Name: Taiz University Country: Yemen Address: P.O.Box 6803, Taiz / Ta'izz Email: President.office@taiz.edu.ye Phone Number: +9674242175 We thank you again for your great efforts in handling and reviewing our manuscript, highly appreciated. Sincerely yours.

**Funding:** The author(s) received no specific funding for this work.

**Competing interests:** The authors have declared that no competing interests exist.

**Abbreviations:** COVID-19, Coronavirus disease 2019; PTSD, post-traumatic stress disorder; IES-R, Impact of Event Scale-Revised; PHQ-9, by Patient Health Questionnaire-9; GAD-7, Generalized Anxiety Disorder; ISI, Insomnia Severity Index; CFA, Confirmatory Factor Analysis; SEM, Structural Equation Modeling; MIMIC, Multiple Indicators and Multiple Causes; DIF, Differential Item Functioning; SARS, Severe Acute Respiratory Syndrome; MERS, Middle East Respiratory Syndrome; DASS-21, Depression, Anxiety and Stress Scale—21; MCMI, Multiple Causes and Multiple Indicators; CFI, Comparative Fit Index; IFI, Incremental Fit Index; TLI, Tucker Lewis Index; SRMR, Standardized Root Mean Residual; RMSEA, the Root Mean Squared Error of Approximation; C.R, critical ratio.

# Introduction

The coronavirus disease 2019 (COVID-19) pandemic continues to be an extreme health emergency [1] because the number of infected people exceeds 600 million confirmed cases of COVID-19, including more than 6 million deaths worldwide up to October 12, 2022 [2]. Moreover, COVID-19 has caused disorder on the world's healthcare infrastructure, daily well-being and routine, business, transportation, lifestyle, freedom of movement, education, distribution of medical properties, and economy [3–10]. For these reasons, it is critical to understand how the population has responded to this significant crisis [9]. In the areas affected by the Severe Acute Respiratory Syndrome (SARS) pandemic, moderate to severe PTSD was observed [11]. Similarly, the impact of Middle East Respiratory Syndrome (MERS), H1N1 influenza (swine flu), and Ebola pandemics on psychological health, such as depression and substance abuse, have been documented [11].

The COVID-19 pandemic provoked psychological distress among the general population [12–18]. It has been reported that 71% out of 4615 participants experienced psychological distress [19]. Furthermore, previous studies on healthcare workers reported that frontline healthcare professionals' anxiety, depression, and secondary traumatization scores on COVID-19 were considerably higher than those of other health professionals or non-medical professionals [20–22]. Moreover, healthcare practitioners, who were afraid of being infected with COVID-19, felt stigmatized and experienced significant levels of anxiety, depressive symptoms, and sleep disorders. In addition, poor sleep quality was linked to high anxiety and depression symptoms in the general population and healthcare workers [23–29]. In line with this, others have reported that academicians have high stress, anxiety, and distress frequency. At the same time, out of 349 physicians, 47.9% reported anxiety symptoms, 60.2% reported distress, 21.8% reported burnout, and 10.6% reported depression symptoms [30].

On the other hand, COVID-19 exacerbated the psychological symptoms in people with mental health problems [31–33]. Psychiatric patients, for example, scored considerably higher on the overall IES-R, DASS-21 depression, anxiety, and stress subscales during the peak of the COVID-19 pandemic with strict lockout measures. More than 25% of psychiatric patients were found to have PTSD-like symptoms with moderate to severe sleeplessness. They had a significantly higher rate of moderate to severe clinical insomnia than healthy controls [33]. In addition, the experience of being isolated may be harmful since the data revealed that many people suffer from various long-term mental health issues [11, 31].

A few studies have been conducted about COVID-19's psychological impact on the general population [17, 33–38]. Furthermore, previous studies have indicated a broad spectrum of psychological effects on people during the COVID-19 pandemic at the individual, community, and worldwide levels [14, 34, 39]. Therefore, there is an imperative need to have high-quality data examining the psychological impacts of the COVID-19 pandemic in the general population [39–41]. Although the conducted studies focused on measuring the effects of the COVID-19 pandemic on psychological health, additional evidence is needed to fill up the gap regarding the applicable measurement scales and instruments to measure post-traumatic stress disorder (PTSD) and some psychological outcomes (e.g., depression, anxiety, and insomnia) of COVID-19 worldwide [5, 6, 15, 42]. As a result, the current study aims to conduct an advanced psychometric analysis to examine the impact of the COVID-19 pandemic on psychological outcomes (depression, anxiety, and insomnia), taking into account the presence or absence of PTSD. Moreover, the current study aims to identify the differences in psychological impacts (depression, anxiety, and insomnia) among participants with and without PTSD as latent factors and items within the Multiple Causes and Multiple Indicators (MCMI) and Differential Item Functioning (DIF) framework.

## Materials and methods

### Study design and setting

A cross-sectional study using an online questionnaire was carried out to measure the psychological response during the COVID-19 pandemic. Based on a covariance matrix analysis, this study fulfilled the descriptive and correlational casualty design criteria. The data of the study was collected from 20 countries through an electronic survey.

### Ethics and consent statement

The Taiz university research ethics committee approved the study with the ethical approval reference number: Taiz/RSCGS/2020/03/26/0236. All participants signed an electronic informed consent that included information on the purpose of the study, the methods, the advantages of participation, the voluntary involvement, and the researchers' contact information. The researchers of this study confirm that all methods related to the human participants were performed following the Declaration of Helsinki.

### Participants and procedure

A total of 999 individuals from 20 countries (Fig 1) participated in this study using convenience sampling. Specifically, the study participants were recruited using social media; WhatsApp, Facebook, and emails from 01/04/2020 to 16/05/2020 using an online survey. Due to the lockdown, movement control order, the spread of COVID-19, and infection control, physical distribution and face-to-face contact were not possible [33, 43]. A Google Form was used to generate an electronic survey sent later via a hyperlink to collect the data. The participants' confidentiality was protected throughout the data collection and analysis to guarantee data integrity. Before the data collection procedures, all participants were briefed via an

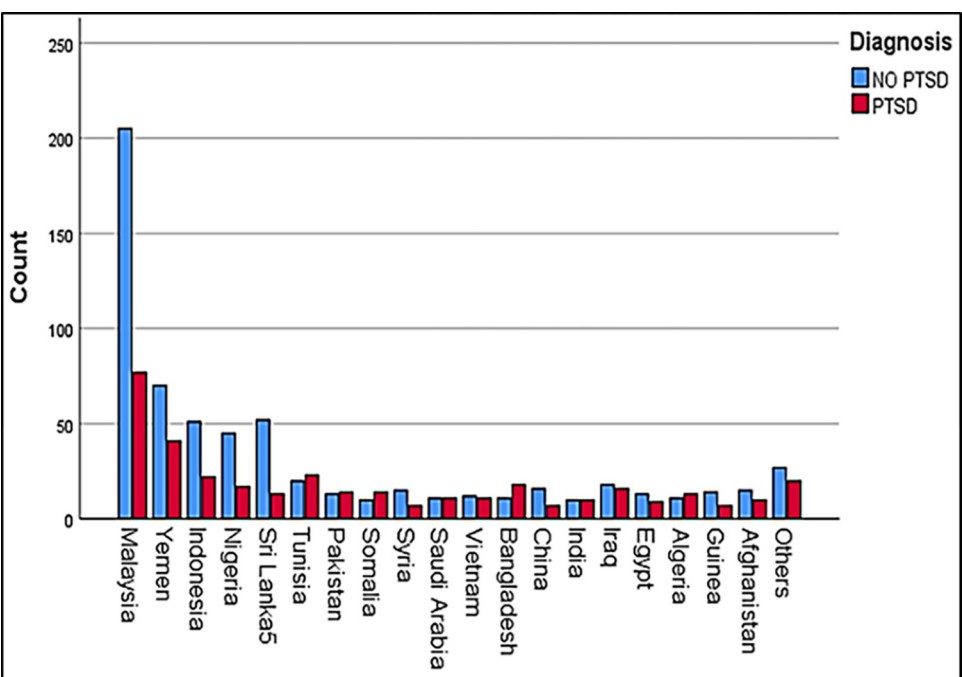

**Fig 1. Lists of countries involved in the present study.**

introduction page of the Google Form about the purpose of the study, data privacy, and its exclusive scientific use, submitted their informed consent, and for any clarification about the questionnaire. Participants were requested to engage in the study willingly (i.e., they had to provide an e-consent before they began the survey), and 999 out of 1020 participants agreed to participate.

## Variables and instruments

The survey was divided into socio-demographic data, and COVID-19-related data, which were specifically constructed was explicitly constructed for this study using a validated and reliable questionnaire that included the IES-R, PHQ-9, GAD-7, and ISI. The Impact of Event Scale-Revised (IES-R) was used to assess PTSD caused by COVID-19. The IES-R is a self-administered questionnaire that has been well-validated in different regions worldwide to measure the level of psychological impact within two weeks of being exposed to a public health crisis. The scale consists of 22 items categorized into three subscales: avoidance, intrusion, and hyperarousal [15, 44]; S4 Table presents IES-R items and symbols for Covid-19. The overall IES-R mean score was classified into four categories; 0–23 (normal), 24–32 (mild psychological impact), 33–36 (moderate psychological impact), and >37 (severe psychological impact) [45]. According to the categories, we further divided the participants into two groups: those without PTSD (score ranging from 0 to 23) and those with PTSD (score more than 23).

The Patient Health Questionnaire-9 (PHQ-9) was used to assess depression. The PHQ-9 is a 9-item self-report instrument for screening, monitoring, diagnosing, and measuring the depression severity of [46]. The PHQ-9's psychometric properties have already been established in Chinese populations [47]. Recent studies by Civantos et al. and Kroenke et al. [30, 46], defined the symptoms of depression as a total score of ≥5 points on the PHQ-9. A cut-off of ≥10 indicated a possible major depression, with a sensitivity of 80.0% and specificity of 92.0%. The participants' response rate options were categorized as follows: 0 = "not at all", 1 = "several days", 2 = "more than half the days" and 3 = "nearly every day". The overall score varied from zero to 27 with a higher score indicating more self-reported depression [48, 49]. The PHQ-9 total scores were grouped into four categories: 0–4 (normal), 5–9 (mild depressive disorder), 10–14 (moderate depressive disorder), 15–19 (moderately severe depressive disorder), and 20–27 (severe depressive disorder) [50].

The 7-item Generalized Anxiety Disorder (GAD-7) Scale was used to evaluate anxiety symptoms during the previous two weeks. The GAD-7 has already been used and proven to be a reliable measure for measuring anxiety symptoms [23, 30, 51, 52]. The GAD-7 Scale scores ranged from 0–21, and a score of 10 was reported to be the endpoint for detecting cases of GAD-7 ≥ 10 (high anxiety). The scores were grouped into four categories: normal (0–4), mild (5–9), moderate (10–14), and severe (15–21) [23, 30].

Insomnia was assessed using the Insomnia Severity Index (ISI) [53, 54]. The 7-item self-report index ranged from 0–28 and was used by the scoring system to evaluate the severity of insomnia as follows: no significant insomnia (0–7), subthreshold insomnia (8–14), moderately severe insomnia (15–21), and severe insomnia (22–28) [33, 43, 55].

## Data processing and analysis

Descriptive statistics were firstly carried out to understand the present sample's demographic characteristics and the scores of the tested constructs, including Impact of Event Scale-Revised (IES-R) for COVID -19 (Intrusion, Avoidance, and Hyperarousal) and Psychological Outcomes (Depression, anxiety, and insomnia). All items were normally distributed, and the

overall reliabilities for each dimension and each item were above average, with 0.70 to reach 0.90 as a slightly excellent degree (S1 and S2 Tables).

Structural Equation Model (SEM) and Confirmatory factor analysis (CFA) were used with their model-data fit assessed using several approximate fit indices, including Comparative Fit Index (CFI), Incremental Fit Index (IFI), Tucker Lewis Index (TLI), Standardized Root Mean Residual (SRMR), and the Root Mean Squared Error of Approximation (RMSEA) with a confidence interval of lower and upper limits. Good model-data fit was based on the RMSEA and CFI values of 0.06 or less and 0.95 or above, respectively [5, 56–59]. The acceptable model-data fit was assessed by RMSEA values of 0.08 and CFI values of 0.90 [60]. In contrast, the perfect model-data fit was assessed by RMSEA values of 0.000 and CFI values of 1.000. SRMR, IFI, and TLI values were conducted with a similar rate to RMSEA and CFI. Statistical significance was tested based on a critical ratio (CR) above 1.964 and a p-value less than 0.05 [61, 62]. Factor loading was preferred at 0.60, and if its value was less than 0.60, it was removed.

Multiple Indicators and Multiple Causes (MIMIC), a particular and essential application of SEM in study validation, allows for the investigation of multi-group variations on a latent construct [63, 64]. According to the membership group interpretation method, a significant positive regression coefficient indicates a better value for one group's specific component. A significant negative regression coefficient more excellent value for one group's component. On the other hand, a significant negative regression coefficient implies a lower value on the specific factor for another group. The measurement set detected Differential Item Functioning (DIF) in observed indicators of latent variables [65].

The Differential Item Functioning (DIF) is used to determine if an assessment has a systematic bias due to construct-irrelevant variables. DIF depicts that the degree of performance on a given item is systematically varied. For example, the DIF is proved when examinees in subgroups (e.g., race/ethnicity, gender, socioeconomic status) with the same level of latent qualities have different probabilities of responding correctly to a particular item [66]. Therefore, the MIMIC confirmatory factor analysis was used in this study to examine DIF in polytomous items, which are commonly used in educational assessments (e.g., constructed-response items) and psychological inventories (e.g., Likert-type items and rating scale items) [67]. In a simple MIMIC model, one latent factor is regressed on an observable grouping variable to allow for group means differences on the factor. An item is assessed for DIF by regressing it (i.e., replies to it) on the grouping variable. If a group association significantly predicts item responses after controlling for group mean differences on the factor, there is evidence of different functioning [68]. The MIMIC method with a pure anchor (denoted as M-PA) was conducted for this research. M-PA means where all items are studied except Item 1, which is the anchored item [67]. When an item on a test has distinct measuring qualities for one group of people versus another, regardless of the group-mean differences on the variable under investigation, this is known as DIF. DIF identification is crucial since it can lead to incorrect assumptions or judgements concerning group variances, as well as invalidate procedures for determining conclusions about an individual. The MIMIC model was also used to examine differences in the scores of the three-factor model of psychological outcomes, depression, anxiety, and insomnia, comparing people with and without PTSD. Different codes denoted group membership: 0 = participants who did not have PTSD (a reference group), and 1 = those who did (a focal group).

## Results

### Demographic results

A total of 999 Participants from 20 countries completed the survey. Malaysia, Yemen, Indonesia, Nigeria, and Sri Lanka had the highest number of participants, followed by other countries

**Table 1. Socio-demographic characteristics of participants (N = 999).**

| | n | % | PTSD Diagnosis | |
|---|---|---|---|---|
| | | | NO | YES |
| **PTSD** | | | | |
| Participants with No PTSD (Reference group) | 639 | 64.0 | - | - |
| Participants with PTSD (Focal group) | 360 | 36.0 | | |
| **Gender** | | | | |
| Female | 554 | 55.5 | 357 (64.4%) | 197 (35.6%) |
| Male | 445 | 44.5 | 282 (63.4%) | 163 (36.6%) |
| **Age** | | | | |
| 18–25 | 242 | 24.2 | 141(58.3%) | 101(41.7%) |
| 26–35 | 403 | 40.3 | 262(65.0%) | 141(35.0%) |
| 36–45 | 254 | 25.4 | 163(64.2%) | 91(35.8%) |
| 46–75 | 100 | 10 | 73 (73.0%) | 27(27.0%) |
| **Marital Status** | | | | |
| Single | 464 | 46.4 | 280 (60.3%) | 184 (39.7%) |
| Married | 496 | 49.6 | 335 (67.5%) | 161 (32.5%) |
| Engaged | 27 | 2.7 | 16 (59.3%) | 11 (40.7%) |
| Divorced | 12 | 1.2 | 8 (66.7%) | 4 (33.3%) |
| **Employment Status** | | | | |
| Students | 551 | 55.2 | 322 (58.4%) | 229(41.6%) |
| Healthcare workers | 53 | 5.3 | 44 (83.0%) | 9 (17.0%) |
| Educational profession | 230 | 23.0 | 157 (68.3%) | 73 (31.7%) |
| Administrative professional | 56 | 5.6 | 40 (71.4%) | 16 (28.6%) |
| Others | 109 | 10.9 | 76 (69.7%) | 33 (30.3%) |
| **Education level** | | | | |
| High school equivalent | 31 | 3.1 | 23 (74.2%) | 8 (25.8%) |
| Bachelor | 285 | 28.5 | 186 (65.3%) | 99 (34.7%) |
| Diploma | 79 | 7.9 | 56 (70.9%) | 23 (29.1%) |
| Master | 367 | 36.7 | 234 (63.8%) | 133 (36.2%) |
| PhD | 237 | 23.7 | 140 (59.1%) | 97(40.9%) |

(Fig 1). In this study, Females (n = 554; 55.5%) were slightly higher than males (n = 445; 45.5%), 35.6% of whom experienced PTSD compared to 36.6% of males. Generally, 64% of the total participants (n = 639) did not have PTSD symptoms. The results revealed that the participants' average age was 33.06, Std. D = 9.3; distributed according to the following age categories: 403 (40.3%) aged 24–35 years and 141 (35.0%) of them experienced PTSD, 254 (25.4%) aged 36–45 years and 91(35.8%) of them diagnosed with PTSD, 242 (24.2%) aged 18–25 years and 101(41.7%) of them experienced PTSD, and 100 (10%) aged 46–75 years and 27(27.0%) of them diagnosed with PTSD. Regarding marital status, 46.4% of participants were single, while 49.6% were married. The majority of the participants (n = 551; 55.2%) were students, followed by participants who work in an educational profession (n = 230; 23%). In terms of education level, the majority (n = 604; 60.5%) of the participants held a postgraduate qualification (master's or Ph.D. degrees) (Table 1).

As shown in S1 Table, the overall mean score of intrusion, avoidance, and hyperarousal dimensions of the IES-R scale were 0.90, 1.07, and 0.91, respectively, with a standard deviation of ±1.045, ±1.15, and ±1.12. Moreover, as shown in Table 2, 639 (64.0%) of the participants experienced a normal level of impact events of COVID-19, whereas the remaining 360 (36.0%) experienced a mild to severe impact. The overall means of depression, anxiety, and insomnia

**Table 2. Levels of the psychological impact of COVID-19, depression, anxiety, and insomnia (N = 999).**

| Levels | Frequency (n) | Percent (%) |
|---|---|---|
| **IES-R Levels** | | |
| 0–23 (normal) | 639 | 64.0 |
| 24–32 (mild psychological impact) | 146 | 14.6 |
| 33–36 (moderate psychological impact) | 41 | 4.1 |
| >37 (severe psychological impact) | 173 | 17.3 |
| **Depression Levels** | | |
| (0–4) No Symptoms | 308 | 30.8 |
| (5–9) Mild Depressive Symptoms | 326 | 32.6 |
| (10–14) Moderate Depressive Symptoms | 179 | 17.9 |
| (15–19) Moderately Severe Depression | 104 | 10.4 |
| (<20) Severe Depression | 82 | 8.2 |
| **Anxiety Levels** | | |
| (0–4) normal | 431 | 43.1 |
| (5–9) mild | 331 | 33.1 |
| (10–14) moderate | 145 | 14.5 |
| (15–21) severe | 92 | 9.2 |
| **Insomnia Levels** | | |
| 0–7 = No clinically significant insomnia | 484 | 48.4 |
| 8–14 = Sub-threshold insomnia | 312 | 31.2 |
| 15–21 = Clinical insomnia (moderate severity) | 162 | 16.2 |
| 22–28 = Clinical insomnia (severe) | 41 | 4.1 |

were 0.86, 0.907, and 1.27, respectively, with a standard deviation of ±0.94, ±0.932, and ±1.13 (S2 Table). As shown in Table 2, 308 (30.8%) reported no depression symptoms among the participants. In comparison, 691 (69.2%) reported mild to severe depression, and 431 (43.1%) out of all participants had no anxiety, while 568 (65.9%) had mild to severe anxiety. Regarding insomnia, the results demonstrated that 484 (48.4%) of the participants reported no clinically significant insomnia, and 515 (51.6%) reported sub-threshold insomnia to severe clinical insomnia (Table 2).

## Psychometric findings

**Impact of Event Scale-Revised (IES-R) for COVID-19.** In the first order for the original model of Impact of Event Scale-Revised for COVID-19 with three dimensions: intrusion with 8 items, avoidance with 8 items, and hyperarousal with 6 items could not achieve the satisfactory fit indices (CFI = 0.853, IFI = 0.854, TLI = 0.835, SRMR = 0.057, RMSEA = 0.092) (Fig 2). After deleting six items for the reasons of non–positive variance, the goodness of fit statistics was reasonable (CFI = 0.935, IFI = 0.935, TLI = 0.923, SRMR = 0.067, RMSEA = 0.070). Three dimensions of IES-R for COVID-19: intrusion, avoidance, and hyperarousal, were statistically significant (T-value ≥ 1.964 and p-value ≤ 0.001). The positive correlation values ranged from 0.97 to 0.77, and the unstandardized estimation for all Impact of Event Scale-Revised items for COVID-19 was statistically significant (T-value ≥ 1.964 and p-value ≤ 0.001). Moreover, the standardized estimate of Factor Loading was equal to or greater than 0.60 as sufficient loading. The sufficient rate of average variance extracted (AVE) was greater than 0.50, and excellent composite reliability coefficients were above 0.70, reflecting the convergent validity of IES-R for COVID-19 (Fig 2 and S1 Table).

**Depression, anxiety, and insomnia.** One item of the depression factor (Item 9) was below 0.60, so it was removed. Subsequently, the goodness of fit statistics was reasonable

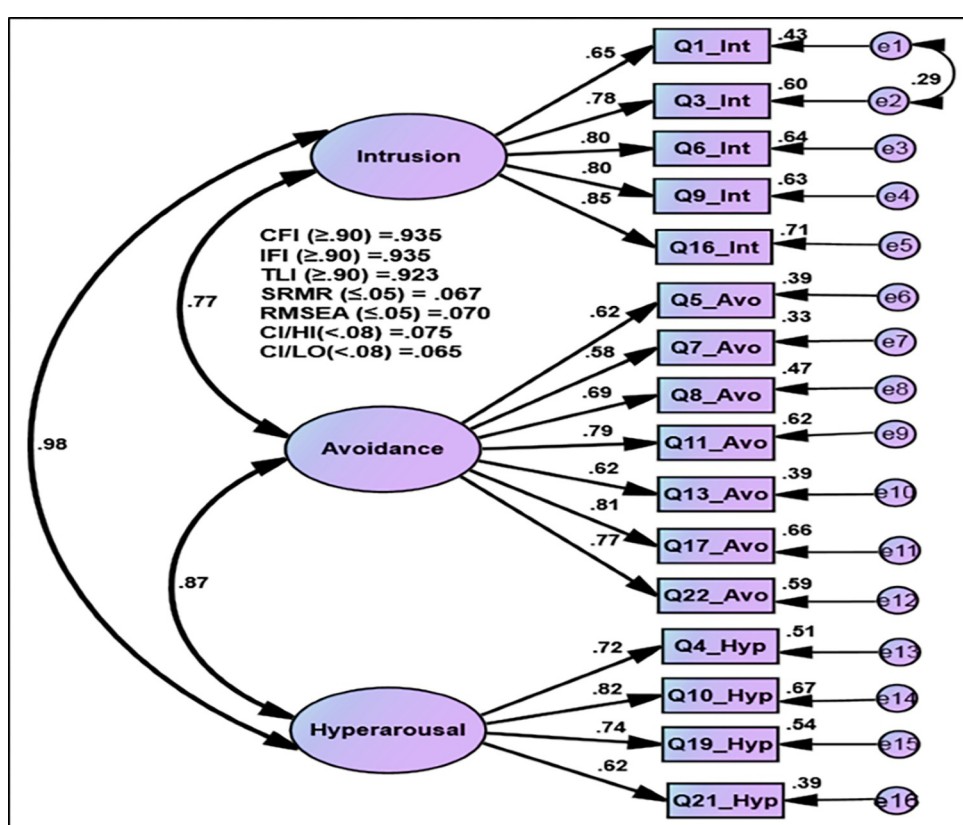

**Fig 2. Confirmatory Factor Analysis (CFA) for IES-R for COVID-19.**

(CFI = 0.929, IFI = 0.929, TLI = 0.920, SRMR = 0.048, RMSEA = 0.067) (Fig 3). The three factors of psychological outcomes: depression, anxiety, and insomnia, and the unstandardized estimation for all items were statistically significant ($T$-value $\geq 1.964$ and $p$-value $\leq 0.001$). The positive values of correlation ranged from 0.79 to 0.61. The standardized estimation of Factor Loading was above 0.60 as sufficient loading (Fig 3 and S1 Table). The sufficient rate of average variance extracted (AVE) was above 0.50 except for depression (0.48), and the excellent coefficients of composite reliability of above 0.70 were obtained, indicating convergent validity of psychological outcomes.

### The effect of IES-R for Covid-19 on psychological outcomes

**Model 1: IES-R for Covid-19 and psychological outcomes.** The mean, standard deviation, skewness, kurtosis, and correlation of hypothesized model of IES-R for COVID-19 and psychological outcomes are illustrated in Table 4. Error terms were correlated between anxiety and insomnia to reach excellent goodness of fit statistics. Subsequently, the goodness of fit statistics were reasonable (CFI ($\geq 0.90$) = 0.995, IFI ($\geq 0.90$) = 0.995, TLI ($\geq 0.90$) = 0.989, SRMR ($\leq 0.08$) = 0.0153, RMSEA ($\leq 0.08$) = 0.051, CI/HI ($<0.08$) = 0.030, and CI/LO ($<0.08$) = 0.073) (Fig 4). The results demonstrated a positive correlation between IES-R for COVID-19 (intrusion, avoidance, and hyperarousal) and psychological outcomes (depression, anxiety, and insomnia). They were statistically significant ($T$-value $\geq 1.964$ and $p$-value $\leq 0.001$, R = 0.71) (Fig 4 and Table 3).

The unstandardized estimation for all dimensions of IES-R for COVID-19 (intrusion, avoidance, and hyperarousal) and psychological outcomes (depression, anxiety, and insomnia)

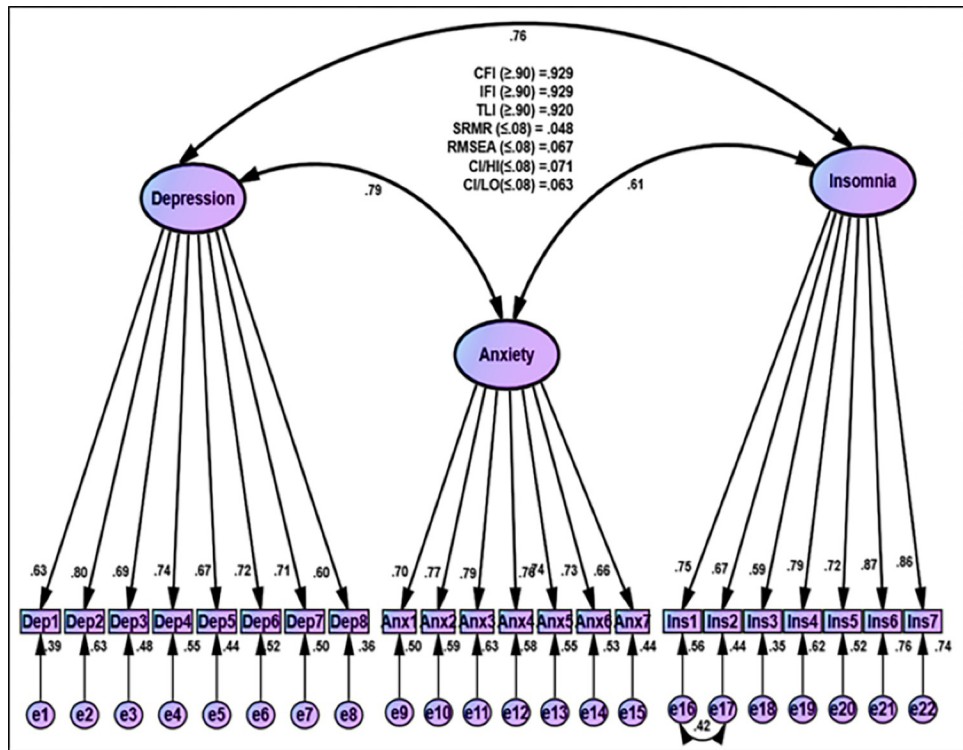

**Fig 3. CFA for psychological outcomes (Depression, anxiety, and insomnia).**

was found to be statistically significant (*T*-value ≥ 1.964 and *p*-value ≤ 0.001). Furthermore, the standardized factor loading estimations are excellent (above 0.70) (Fig 4 and Table 4).

**Model 2: Structural model for the effect of IES-R for Covid-19 on psychological outcome.** The results showed that the IES-R for COVID-19 (intrusion, avoidance, and hyperarousal) was positively associated with psychological outcomes (depression, anxiety, and insomnia) and this hypothesis was statistically significant (b = 1.114, SE = 0.054, *T*-value = 20.556 (≥ 1.964) and *p*-value = 0.001 (≤ 0.001), β = 0.715 and η = 0.51) (Fig 4).

Fig 5 presents the hypothesized model of IES-R for COVID-19 as individual factors (intrusion, avoidance, and hyperarousal) and psychological outcomes as individual factors

**Table 4. Parameters of IES-R for Covid-19 and psychological outcomes.**

| Latent Variables | Factor | B | SE | CR | p | λ | SMC |
|---|---|---|---|---|---|---|---|
| **IES-R for Covid-19** | Intrusion | 1.000 | - | - | - | 0.838 | 0.702 |
| | Avoidance | 1.219 | 0.045 | 27.110 | *** | 0.750 | 0.563 |
| | Hyper-arousal | 0.843 | 0.024 | 34.703 | *** | 0.933 | 0.870 |
| **Psychological Outcome** | Depression | 1.000 | - | - | - | 0.852 | 0.726 |
| | Anxiety | 0.845 | 0.032 | 26.169 | *** | 0.841 | 0.707 |
| | Insomnia | 0.923 | 0.036 | 25.426 | *** | 0.821 | 0.675 |
| **Correlation** | F1-F2 | 14.522 | 0.965 | 15.050 | *** | 0.715 | 0.511 |
| | e5-e6 | -4.421 | 0.729 | -6.064 | *** | -0.400 | 0.160 |

B = unstandardized estimates, S.E = Stander Error, C.R = Critical Ratio, P = probability, λ = loading, SMC = Squared Multiple Correlation, F1 = IES-R for Covid-19, F2 = Psychological Outcome

*** = significance at 0.001.

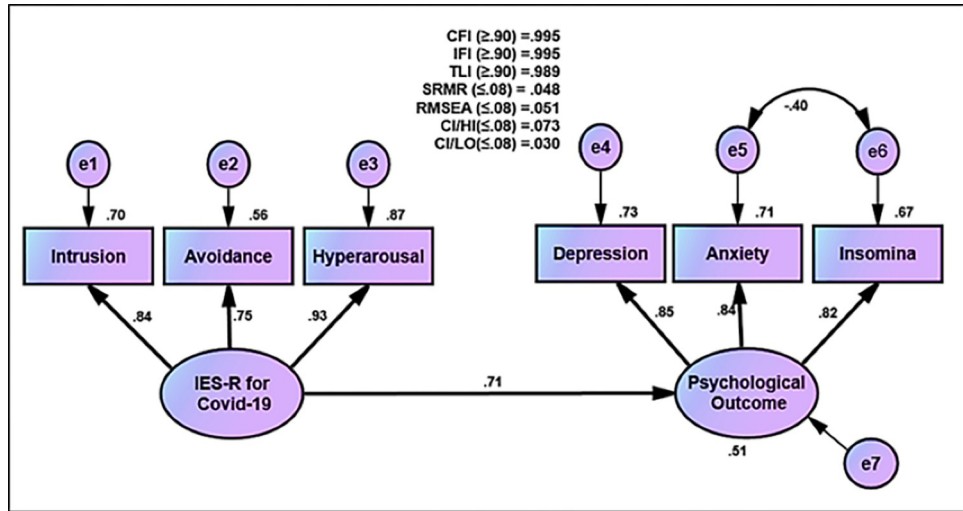

**Fig 4. Effect of IES-R for Covid-19 on psychological outcomes.**

(depression, anxiety, and insomnia), which achieved perfect fit goodness statistics after correlating exogenous and endogenous factors error terms. The goodness of fit statistics was highly reasonable (CFI ($\geq$0.90) = 0.1000, IFI ($\geq$0.90) = 1.000, TLI ($\geq$0.90) = 1.004, SRMR ($\leq$0.08) = 0.001, RMSEA ($\leq$0.08) = 0.000, CI/HI ($<$0.08) = 0.032, and CI/LO ($<$0.08) = 0.000) (Fig 5). The results demonstrated that the intrusion, avoidance, and hyperarousal significantly and positively affected psychological outcomes, depression, anxiety, and insomnia. As a result, people who experienced intrusion/ avoidance/ hyperarousal as the impact of COVID-19 would be subjected to depression, anxiety, and insomnia with effect sizes of 0.34, 0.36, and 0.30, respectively, with a large effect size (above 0.25) (Fig 5 and Table 5).

**Differences between psychological distress (depression, anxiety, and insomnia) between groups with and without PTSD.** A path diagram of the MIMIC model results of the three-factor model of psychological outcomes is given in Fig 6. Correlated error terms among the three-factor model were conducted based on essential requirements of the MIMIC model. Goodness of Fit indices in the proposed MIMIC model were acceptable (CFI = 0.929, IFI = 0.920, TLI = 0.929, SRMR = 0.046, RMSEA = 0.065) (Figs 6 and 7).

The results in the mean structure case are reflected in the regression of the latent three factors on group differences. Negative coefficients indicated that participants without PTSD scored lower and had a negative effect on depression, anxiety, and insomnia compared to participants with PTSD, who scored higher with positive coefficients (Fig 7 and Table 6). Table 6

**Table 3. Descriptive statistics and correlation matrix for IES-R for Covid-19 and psychological outcomes (Depression, anxiety, and insomnia).**

|  | 1 | 2 | 3 | 4 | 5 | 6 | Skewness | Kurtosis |
|---|---|---|---|---|---|---|---|---|
| **Intrusion** | 1.00 |  |  |  |  |  | 1.029 | 0.794 |
| **Avoidance** | 0.623 | 1.00 |  |  |  |  | 1.175 | 1.150 |
| **Hyperarousal** | 0.779 | 0.705 | 1.00 |  |  |  | 0.804 | 0.060 |
| **Depression** | 0.513 | 0.451 | 0.568 | 1.00 |  |  | 0.736 | -0.122 |
| **Anxiety** | 0.560 | 0.424 | 0.566 | 0.71 | 1.00 |  | 0.868 | 0.276 |
| **Insomnia** | 0.475 | 0.425 | 0.534 | 0.70 | 0.567 | 1.00 | 0.736 | -0.230 |
| **Mean** | 5.07 | 3.35 | 7.36 | 8.07 | 8.63 | 8.93 |  |  |
| **Std. Deviation** | 4.31 | 3.26 | 5.86 | 5.65 | 6.60 | 6.32 |  |  |

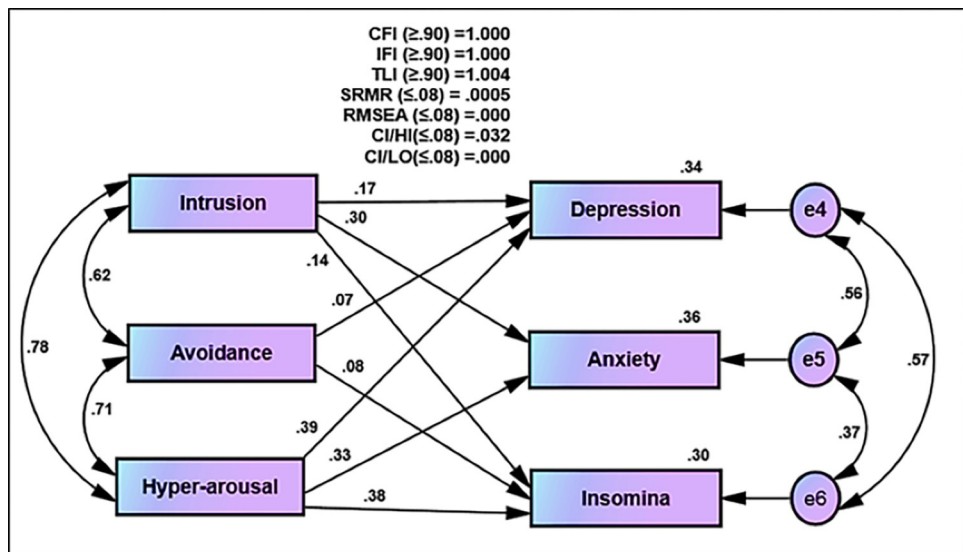

**Fig 5. Effect of IES-R for Covid-19 on psychological outcomes (individual factors).**

shows the unstandardized and standardized coefficients, as well as inferential statistics (i.e., T-values and P-values). The findings indicated that participants without PTSD were negatively rated and scored lower, on the mentioned factors, than those who experienced PTSD.

Moreover, to analyze the equality and differences between participants with and without PTSD groups on the score of each item of the three-factor model of the psychological outcomes, several paths via arrows were linked from the membership group to each item, with the exception of fixed items from latent variable to that item (e.g., Dep1, Anx1, and Ins1) based on procedures of DIF (Figs 8 and 9). These procedures were performed within the framework of the MIMIC model. Goodness of fit indices in MIMIC model obtained plausible rate (CFI = 0.932, IFI = 0.932, TLI = 0.916, SRMR = 0.044, RMSEA = 0.066) (Figs 8 and 9). Regarding the Structural Component of MIMIC of DIF, participants without PTSD obtained negatively lower scores (Fig 8 and Table 7). In contrast, participants with the PTSD group in terms of latent traits of depression, anxiety, and insomnia had higher scores with a positive direction (Fig 9 and Table 7).

**Table 5. Parameters of impact of Covid-19 based on IES-R on individual psychological outcomes.**

| Individual psychological outcomes | IES-R for Covid-19 | B | SE | CR | p | λ | η |
|---|---|---|---|---|---|---|---|
| Depression | Intrusion | 0.255 | 0.063 | 4.014 | *** | 0.166 | 0.34 |
| Depression | Avoidance | 0.082 | 0.034 | 2.396 | 0.017 | 0.073 | |
| Depression | Hyperarousal | 0.783 | 0.090 | 8.715 | *** | 0.387 | |
| Anxiety | Intrusion | 0.398 | 0.053 | 7.487 | *** | 0.303 | 0.36 |
| Anxiety | Hyperarousal | 0.572 | 0.070 | 8.152 | *** | 0.330 | |
| Insomnia | Intrusion | 0.199 | 0.063 | 3.168 | 0.002 | 0.136 | 0.30 |
| Insomnia | Avoidance | 0.081 | 0.038 | 2.125 | 0.034 | 0.075 | |
| Insomnia | Hyperarousal | 0.728 | 0.090 | 8.043 | *** | 0.376 | |

B = unstandardized estimates, S.E = Stander Error, C.R = Critical Ratio, P = probability, λ = loading, η = Effect Size

*** = significance at 0.001

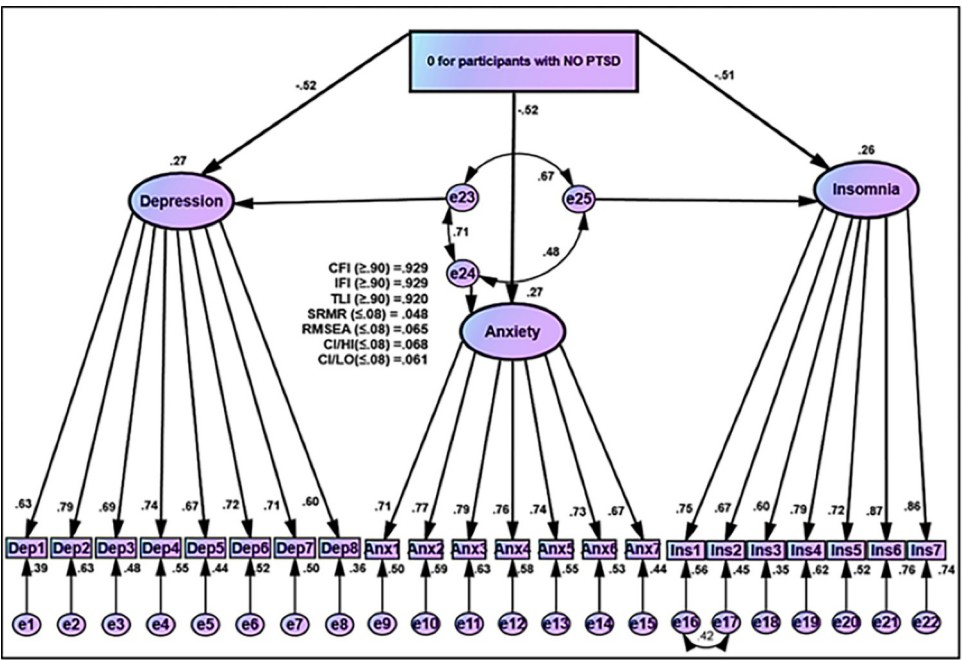

**Fig 6. MIMIC model for participants without PTSD based on IES-R for Covid-19 on psychological outcomes (Depression, anxiety, and insomnia).**

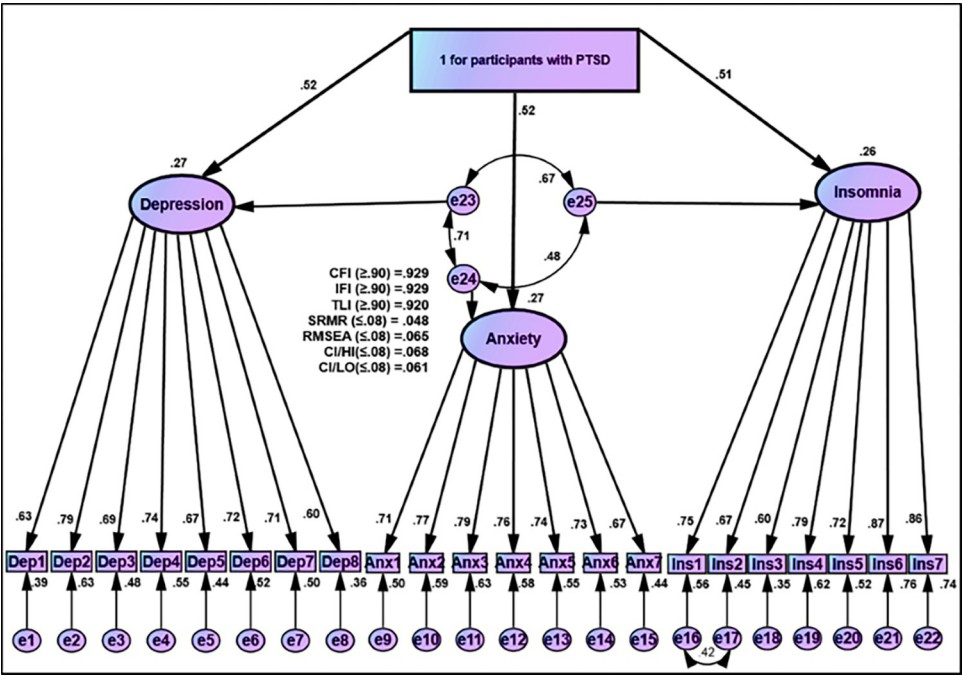

**Fig 7. MIMIC model for participants with PTSD based on IES-R for Covid-19 on psychological outcomes (Depression, anxiety, and insomnia).**

**Table 6. Parameters of Multiple Indicators and Multiple Causes (MIMIC) model.**

| | | | Unstandardized coefficient | S.E | T | Standardized coefficient | P |
|---|---|---|---|---|---|---|---|
| H₁ | Depression | No PTSD | -0.677 | 0.047 | -14.435 | 0.518 | 0.27 |
| | | PTSD | 0.677 | 0.047 | 14.435 | 0.518 | 0.27 |
| H₂ | Anxiety | No PTSD | -0.750 | 0.049 | -15.390 | 0.518 | 0.27 |
| | | PTSD | 0.750 | 0.049 | 15.390 | 0.518 | 0.27 |
| H₃ | Insomnia | No PTSD | -0.950 | 0.059 | -16.144 | 0.509 | 0.26 |
| | | PTSD | 0.950 | 0.059 | 16.144 | 0.509 | 0.26 |

B = unstandardized estimates, S.E = Standard Error

## Discussion

Several pieces of data indicated that the COVID-19 pandemic has had substantial psychological consequences. According to recent research, the COVID-19 pandemic is associated with distress, anxiety, sadness, and insomnia in the general population worldwide [18]. In addition, the mental health sequelae of the pandemic are likely to last for months or even years and might peak later than the time of the actual pandemic outbreak. Thus, further research is needed to determine how the mental health consequences of the COVID-19 pandemic can be reduced during and after the outbreak [10, 18, 19, 22, 26, 28, 41, 50, 55]. Therefore, this study aimed to investigate the psychological impact of the COVID-19 pandemic on the overall population in multi-countries during its initial phase.

Our socio-demographic data indicated that participants without PTSD rated higher (n = 639; 64%) than participants with PTSD (n = 360; 36.0%), which means that the PSTD symptoms of the COVID-19 pandemic were lower than those recently reported by Pazmino Erazo et al. which displayed a higher rate of PSTD symptoms (43.8%) [27]. In addition, the 22-item IES-R scoring diagnosis of PTSD found that males rated (36.6%) higher than females

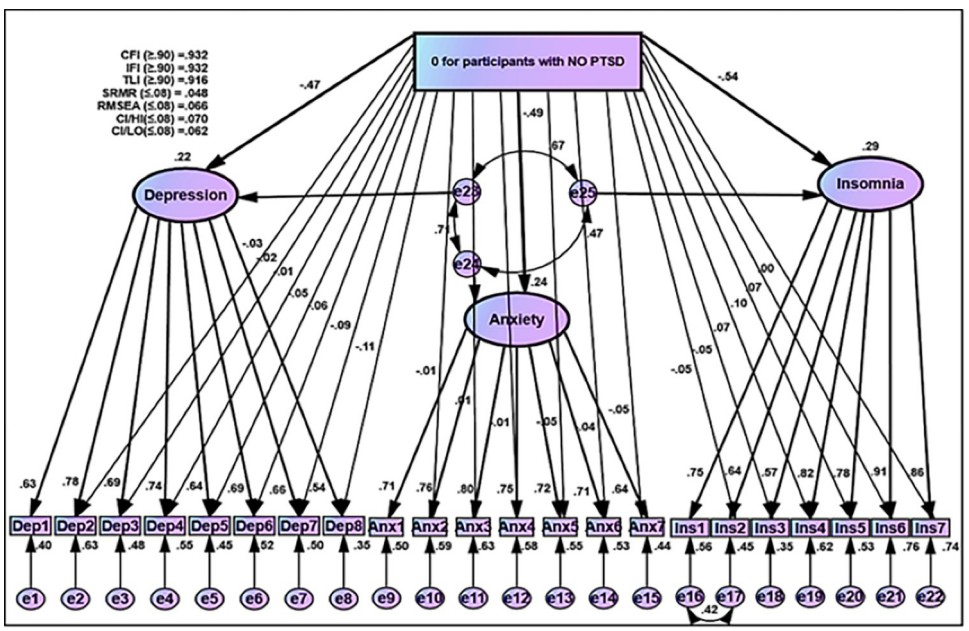

**Fig 8. MIMIC for DIF for participants without PTSD based on IES-R for Covid-19.**

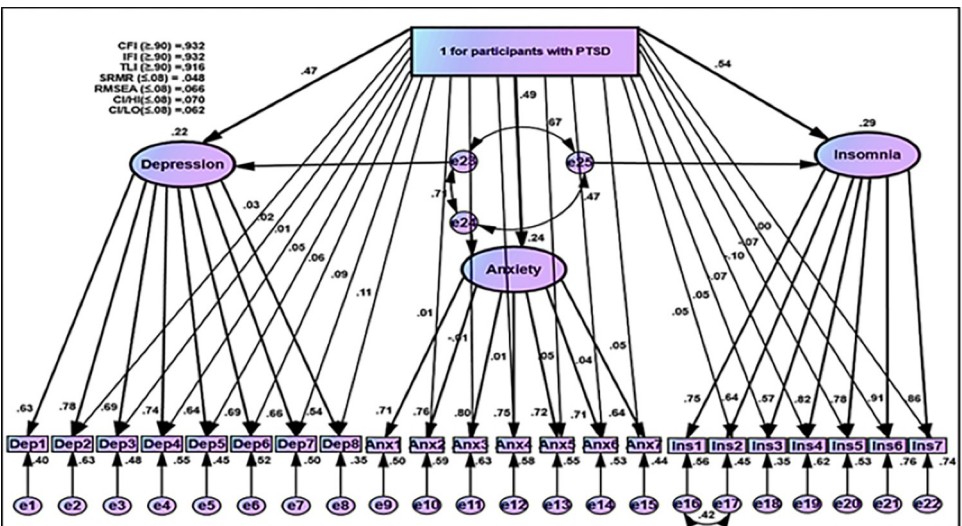

**Fig 9. MIMIC for DIF for participants with PTSD based on IES-R for Covid-19.**

(35.6%). Contrary to our findings, Wang et al. reported that the pandemic's psychological impact on females was more significant than on males [34]. Accordingly, the results revealed that group scores of participants without PTSD (Fig 6) were less negative in terms of the psychological effects such as insomnia, depression, and anxiety than participants with PTSD who scored high with positive coefficients (Fig 7). In consistent with this, the findings of the current study indicated that participants with PTSD scored higher in depression, anxiety, and insomnia than those without PTSD. The current study findings are in line with a previous study conducted by Janati Idrissi et al., who reported that 35.6% of participants had symptoms of depression [28]. However, Choi et al. found that only 19% of respondents had depression (PHQ-9 score $\geq$ 10), presenting approximately half of the percentages found in the current study [50]. Also, Alkhamees et al. revealed that 16.4% of participants had severe depressive symptoms, slightly less than half of the percentage found in the current study [18]. Contrary to this, Elhadi et al. reported a higher rate of depression symptoms during the COVID-19 pandemic. The authors indicated that 46.2% of participants experienced symptoms of depression [26].

Concerning anxiety, the results of the current study found that 235 respondents (23.7%) (GAD-7 score $\geq$ 10) had moderate to severe symptoms of anxiety, whereas 331 (33.1%) had mild symptoms of anxiety. However, 431 (43.1%) participants did not experience any anxiety

**Table 7. Parameters of structural component of MIMIC for DIF.**

|  | Group | B | S.E | T | p | λ |
|---|---|---|---|---|---|---|
| **Depression** | **No PTSD** | -0.622 | 0.063 | -9.855 | *** | -0.473 |
|  | PTSD | 0.622 | 0.063 | 9.855 | *** | 0.473 |
| **Anxiety** | **No PTSD** | -0.722 | 0.061 | -11.791 | *** | -0.495 |
|  | PTSD | 0.722 | 0.061 | 11.791 | *** | 0.495 |
| **Insomnia** | **No PTSD** | -0.964 | 0.070 | -13.853 | *** | -0.537 |
|  | **PTSD** | 0.964 | O.070 | 13.853 | *** | 0.537 |

B = unstandardized estimates, S.E = Stander Error, C.R = Critical Ratio, P = probability, λ = loading

*** = significance at 0.001

symptoms. These results indicated that approximately a quarter of the study participants suffered from anxiety symptoms during the COVID-19 pandemic associated with PTSD. In comparison with previous reports, 29.5% of participants experienced anxiety symptoms [28], 14% of respondents had anxiety [50], 13.9% of responses indicated anxiety symptoms [18], and 19% of respondents suffered from anxiety symptoms [26].

Insomnia was one of the psychological impacts of the COVID-19 pandemic. The results of the current study found that 515 (51.5%) of the participants experienced symptoms of insomnia based on the Insomnia Severity Index (ISI) (total score ≥ 8). In addition, 484 (48.4%) of participants reported that they did not experience insomnia symptoms during the pandemic. These results revealed that more than half of the participants who answered the questionnaire had insomnia symptoms. Likewise, Janati Idrissi et al. [28] reported that 56.0% of participants suffered from insomnia symptoms, whereas Zhang et al. [29] inferred that the prevalence rate of insomnia among respondents was 36.1%. The highest prevalence of insomnia suggested that the COVID-19 pandemic substantially impacted the psychological well-being and lifestyle of communities both during and after the pandemic. These results, therefore, supported the association between insomnia, depression, and anxiety.

In groups with PTSD and without PTSD on psychological outcomes as items, DIF results indicated psychological outcomes work equivalently with both groups. Hence, both groups of participants perceived items of psychological outcomes equivalently. These apply to all psychological outcomes (PHQ-9, GAD-7, and ISI), reflecting measurement invariance and three scales' validity. The identical results across the two groups demonstrated that the psychological outcome measurement tools were reliable, well-constructed by professionals, and consistent with earlier research [46, 52, 54].

Two items from PHQ-9 (item7 and 8) and two items from ISI (items 5 and 6) displayed DIF. Differentially functional objects might cause measurement bias; thus, they should be removed or modeled as if they were given to various groups separately. Notably, one apparent technique for eliminating DIF is to amend or remove DIF items from existing scales and test for DIF regularly when new measures are created. Another option for dealing with DIF is to simulate it [68].

## Strengths and limitations

This research has the following strengths; the IES-R for COVID-19 is an essential tool for screening distress or traumatic event. validity and utilization of the IES-R and COVID-19 impact are two important contributions. Moreover, the result of the current study provides evidence for the validity of psychological outcomes, particularly those related to depression (PHQ-9), anxiety (GAD-7), and insomnia (ISI). In cross-disciplinary disciplines like mental health, psychology, public health, and medicine, the validation of this research instrument is essential.

Although this research has strengths, it has various limitations: 1) Although participants in this study came from a variety of socio-demographic backgrounds from different countries, the survey items were only administered in English. As a result, some respondents might not have a sufficient level of proficiency in English language to complete the survey. To make sure that the survey questions could be read, we piloted the research. The participants were also told that they could contact the researchers for more information when needed. However, we acknowledged that utilizing psychological tools across cultures can be difficult. Therefore, to prevent any response bias caused by the language barrier in future research, the instruments could be translated into/adapted for use in various languages. 2) Only an online survey was used to collect the data; neither structured nor semi-structured interviews could be conducted

because of the lockdown, the movement control order, and the spread of COVID-19. As a result, it is impossible to guarantee that the participants will take the online survey seriously. Therefore, it is advised that future research use online interviews to make sure that participants pay attention to filling out the survey. Alternatively, some quality control items might be used to verify if participants pay attention to filling out the online survey. 3) this study was performed entirely online. Due to the long working hours, social segregation, and health care professionals' regulations, the cross-sectional design was used. Implementing a longitudinal design to examine the associations between variables under study is suggested for future studies. 4) the results of the current study may not be generalizable due to a lack of a robust sampling frame [39]. However, the researchers opted to conduct the research believing that it is critical to document the secondary traumatization that healthcare providers and the general public went through the COVID phase.

According to the pandemic's trajectory, healthcare professionals' mental health problems may worsen or improve over time. Therefore, further investigation into the long-term psychological repercussions of this group is recommended. This study does not address a number of demographic variables, psychological outcome levels (such as normal, low, moderate, and high), multi-country variables, and other aspects. To test DIF, for instance, the level of gender can be assessed. From the practical perspective, uniform DIF and nonuniform DIF are arguably the most straightforward to perceive within the MIMIC framework; nevertheless, these issues are not extensively covered in the current study, which is regarded as one of the limitations.

## Conclusion

According to the results of our study, which included 999 participants from various nations, the IES-R for COVID-19 is strongly associated with psychological outcomes (such as depression, anxiety, and insomnia) with a large effect size. Participants with PTSD performed better than those without it in terms of depression, anxiety, and insomnia. Given the results of the current study, government policymakers and healthcare professionals should be informed of the potential dangers of experiencing psychological health issues during COVID-19. The results of the current study, however, cannot establish a causal association because it used a cross-sectional methodology. Future research, therefore, should focus on identifying the causes of the psychological health issues that COVID-19 participants experience and explore potential treatment options. The conduct of experimental research on hospital sampling with COVID-19 patients or survivors is also suggested for future study.

## Supporting information

**S1 Checklist. STROBE statement—Checklist of items that should be included in reports of *cross-sectional studies*.**
(DOCX)

**S1 Table. Descriptive and confirmatory statistics for events.**
(DOC)

**S2 Table. Descriptive and confirmatory statistics for psychological outcomes.**
(DOC)

**S3 Table. Results of Differential Item Functioning (DIF).**
(DOC)

**S4 Table. Items of IES-R for Covid-19.**
(DOC)

## Author Contributions

**Conceptualization:** Musheer A. Aljaberi, Naser A. Alareqe, Abdulsamad Alsalahi, Md. Uzir Hossain Uzir, Atiyeh M. Abdallah, Rukman Awang Hamat.

**Data curation:** Musheer A. Aljaberi, Mousa A. Qasem, Sarah Noman, Md. Uzir Hossain Uzir, Lubna Ali Mohammed, Zine.El.Abiddine Fares.

**Formal analysis:** Musheer A. Aljaberi, Naser A. Alareqe, Chung-Ying Lin.

**Investigation:** Musheer A. Aljaberi, Zine.El.Abiddine Fares, Chung-Ying Lin, Atiyeh M. Abdallah, Rukman Awang Hamat, Mohd Dzulkhairi Mohd Rani.

**Methodology:** Musheer A. Aljaberi, Naser A. Alareqe, Abdulsamad Alsalahi, Md. Uzir Hossain Uzir, Atiyeh M. Abdallah, Rukman Awang Hamat.

**Resources:** Lubna Ali Mohammed, Atiyeh M. Abdallah.

**Software:** Naser A. Alareqe, Chung-Ying Lin.

**Supervision:** Musheer A. Aljaberi.

**Validation:** Musheer A. Aljaberi, Md. Uzir Hossain Uzir, Chung-Ying Lin, Rukman Awang Hamat.

**Visualization:** Musheer A. Aljaberi, Chung-Ying Lin, Rukman Awang Hamat, Mohd Dzulkhairi Mohd Rani.

**Writing – original draft:** Musheer A. Aljaberi, Naser A. Alareqe, Abdulsamad Alsalahi, Mousa A. Qasem, Sarah Noman, Lubna Ali Mohammed, Atiyeh M. Abdallah, Rukman Awang Hamat.

**Writing – review & editing:** Musheer A. Aljaberi, Zine.El.Abiddine Fares, Chung-Ying Lin, Rukman Awang Hamat, Mohd Dzulkhairi Mohd Rani.

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
