## [Decision Letter · Decision Letter 0]

19 Jul 2022

PONE-D-22-16356Impact of COVID-19 Pandemic on the Psychological Outcomes: Multiple Indicators and Multiple Causes (MIMIC) ModelingPLOS ONE

Dear,

Thank you for submitting your manuscript to PLOS ONE. After careful consideration, we feel that it has merit but does not fully meet PLOS ONE’s publication criteria as it currently stands. Therefore, we invite you to submit a revised version of the manuscript that addresses the points raised during the review process. Please submit your revised manuscript by1st September 2022. If you will need more time than this to complete your revisions, please reply to this message or contact the journal office at plosone@plos.org. Please include the following items when submitting your revised manuscript:A rebuttal letter that responds to each point raised by the academic editor and reviewer(s). You should upload this letter as a separate file labeled 'Response to Reviewers'.A marked-up copy of your manuscript that highlights changes made to the original version. You should upload this as a separate file labeled 'Revised Manuscript with Track Changes'.An unmarked version of your revised paper without tracked changes. You should upload this as a separate file labeled 'Manuscript'.

We look forward to receiving your revised manuscript.

Kind regards,

Muhammad Shahzad Aslam, Ph.D.,M.Phil., Pharm-D

Academic Editor

PLOS ONE

Journal Requirements:

a) Did participants provide their written or verbal informed consent to participate in this study?

Reviewers' comments:

Reviewer's Responses to Questions

**Comments to the Author**

1. Is the manuscript technically sound, and do the data support the conclusions?

Reviewer #1: Partly

Reviewer #2: Yes

2. Has the statistical analysis been performed appropriately and rigorously? 

Reviewer #1: Yes

Reviewer #2: Yes

3. Have the authors made all data underlying the findings in their manuscript fully available?

Reviewer #1: No

Reviewer #2: No

4. Is the manuscript presented in an intelligible fashion and written in standard English?

Reviewer #1: Yes

Reviewer #2: Yes

5. Review Comments to the Author

Reviewer #1: Many thanks for giving me the opportunity to review. It is an interesting article where authors studied the psychological impact of COVID-19 pandemic in a multi-country setting. There are the following points that need to be addressed before acceptance of the manuscript.

Methods:

1. Please include the exact duration and time when the study was conducted.

2. It seems that the study participants had diverse socio-demographic characteristics, please also include the age group to add more clarity

3. It is mentioned that participants were recruited via social media, please explain what types of social media sources were used. The Procedure section is missing. I believe such information can be included in it.

4. This study was conducted in a muti-country setting with people of diverse socio-demographic characteristics. It would be helpful if the authors could explain if they translated/adapted the instruments in different languages and how did they conduct that process.

5. Moreover, using psychological instruments cross-culturally is bit challenging, it would be helpful if they could mention the way they managed response bias due to language barrier.

Reviewer #2: Introduction: Well written including references to earlier work related to the area under discussion and express the importance and limitations of what is previously acknowledged.

Method: Full details on how the study was actually carried out is mentioned.

Result: Clearly reveals what actually occurred to the subjects. The results explain the statistical analysis shown in related tables, diagrams and graphs.

Discussion: It includes an absolute comparison of what is already identified in the topic of interest and the clinical relevance of what has been newly established. A discussion on a possible related limitations and necessitation for further studies should also be indicated. Conclusions should ensure that recommendations stated are suitable for the results attained within the capacity of the study. The authors should also concentrate on the limitations in the study and their effects on the outcomes and the proposed suggestions for future studies

6. PLOS authors have the option to publish the peer review history of their article (what does this mean?). If published, this will include your full peer review and any attached files.

Reviewer #1: **Yes: **Bushra Khan

Reviewer #2: **Yes: **Alishba Hania

---

## [Author Response · Author response to Decision Letter 0]

22 Sep 2022

Editor comments and Journal Requirements:

Response:

Thank you very much for your time and consideration in handling and checking our manuscript, and we greatly appreciate it. We have checked PLOS ONE's style requirements, including those for file naming, PLOS ONE style templates and we follow it in the current revised manuscript. 

a) Did participants provide their written or verbal informed consent to participate in this study?

Response: 

Thank you very much for your comment, we have added the Ethics and consent statement section to the methodology on page 5; lines 96-101, and as stated we obtained an electronic informed consent via a Google form from all participants who agree to participate in this study. We obtained an electronic informed consent via a Google form because the contact was not feasible due to the lockdown, movement control order, the spread of COVID 19, and as a part of the infection control during the data collection of this study which was online also due to the same reasons.

3. We note that you have indicated that data from this study are available upon request. PLOS only allows data to be available upon request if there are legal or ethical restrictions on sharing data publicly. For information on unacceptable data access restrictions, please see http://journals.plos.org/plosone/s/data-availability#locunacceptable-data-access-restrictions. In your revised cover letter, please address the following prompts:

Response: 

Thank you very much for your note, yes we indicated that data from this study are available upon request for academic and research purposes because we have stated in the consent form for the participants that their response is confidential and will protect throughout the data collection and analysis to guarantee data integrity and privacy. We have stated also the data is for exclusive scientific use and will not be revealed to any third party only for research and academic purpose upon any request for that. We have provided the contact information for the data access request for any research and academic purpose in our cover letter for submitting our revised manuscript.

Response: 

Thank you very much for your comment, we have moved, and included the ethics statement in the Methods section and deleted it from the sections of the declaration

Reviewer 1:

Many thanks for giving me the opportunity to review. It is an interesting article where authors studied the psychological impact of COVID-19 pandemic in a multi-country setting. There are the following points that need to be addressed before acceptance of the manuscript.

Response: 

Thank you very much Dr. Bushra Khan for your time and great efforts in reviewing our manuscript and for your positive feedback, valuable suggestions, comments, and recommendations to improve the quality of our paper. We have addressed all the concerns raised in the current revised manuscript and this letter; our point-by-point responses will be presented.

Method:

1- Please include the exact duration and time when the study was conducted.

Response: 

Thank you very much for raising this point, we have added the exact date for the data collection from 01/04/2020 to 16/05/2020 in the Participants and Procedure section under the Methodology Page 5; line 105.

2- It seems that the study participants had diverse socio-demographic characteristics, please also include the age

group to add more clarity.

Response: 

We appreciate your efforts in raising this important point for more clarity, we have added the results of participants' age group to the Demographic characteristics results on Page 8&9; lines 193-198, and to table 1 for Socio-demographic characteristics of participants on page 9. The results revealed that the mean age of the participants was 33.06, Std. D= 9.3; distributed according to the following age categories 403(40.3%) their age was between 24-35 years and 141(35.0%) of them experienced PTSD, 254 (25.4%) aged 36-45 years, and 91(35.8%) of them diagnosed with PTSD, following by 242 (24.2%) of the participants aged 18-25 and 101(41.7%) of them experienced PTSD, and 100 (10%) their age was between 46-75 years old and 27(27.0%) of them diagnosed with PTSD.

3- It is mentioned that participants were recruited via social media, please explain what types of social media sources were used. The Procedure section is missing. I believe such information can be included in it.

Response: 

Thank you for raising this point to make the section more informative, We have added the type of social media sources we used; Whats App, Facebook, and emails to recruit the participants as the face-to-face contact and physical distribution of the questionnaire were not feasible due to the lockdown, movement control order, the spread of COVID 19, and as a part of the infection control during the data collection time for the current study. We have also added the Procedure section under the methodology named Participants and Procedure including more information under it regarding the participants and procedure as you suggested, Page 5; lines 102-112.

4&5- This study was conducted in a multi-country setting with people of diverse socio-demographic characteristics. It would be helpful if the authors could explain if they translated/adapted the instruments in different languages and how did they conduct that process. Moreover, using psychological instruments cross-culturally is bit challenging, it would be helpful if they could mention the way they managed response bias due to the language barrier.

Response:

Thank you very much for your comment, we have distributed the questionnaire in English regardless of participants’ country background. Thus, we did not translate or adapt the survey question in different languages. We conducted a pilot study before the actual data collection to make sure of the clarity and readability of the questionnaire items. The pilot study helped us ensure that the item descriptions could be understood by the target participants. Moreover, we provided clear instructions on the introduction page of the Google Form survey and in each section of the questionnaire regarding the answer/fill-in of the survey and we provide them with researchers’ contact details for any clarification or help if they face any difficulties. However, we acknowledged that the participants might have different levels of English and thus we have added it as one of the limitations of our study and as a recommendation for future research on page 17, lines; 382-389.

Reviewer 2:

Introduction: Introduction study and the relevance of the findings. The introduction is a little short and would benefit from some rephrasing and restructuring. In addition, the hypotheses should be stated in the introduction instead of in the results.

Response:

Thank you very much Dr. Alishba Hania for your efforts and positive feedback. we appreciate how thoroughly you reviewed and checked the manuscript sections.

Method: Full details on how the study was actually carried out is mentioned.

Response:

Thank you for your efforts and positive feedback.

Results: Clearly reveals what actually occurred to the subjects. The results explain the statistical analysis shown in related tables, diagrams and graphs.

Response:

Thank you very much for your positive feedback.

Discussion: It includes an absolute comparison of what is already identified in the topic of interest and the clinical relevance of what has been newly established. A discussion on a possible related limitations and necessitation for further studies should also be indicated. Conclusions should ensure that recommendations stated are suitable for the results attained within the capacity of the study. The authors should also concentrate on the limitations in the study and their effects on the outcomes and the proposed suggestions for future studies.

Response:

Thank you for the guidance. We have now elaborated on how future studies can be conducted with the consideration of the present study’s limitations, pages 18&19; lines 382-398. Regarding the Conclusions section, we have toned down our statement to fit the capacity of the present study, page 19; lines 410-419. 

We thank you once again for your immense efforts and positive feedback.

*We also attached a comprehensive response to the editor and reviewers' comments, and Journal requirements presented in table form in reviewers response letter point by point as attached in the document.

Sincerely yours,

---

## [Decision Letter · Decision Letter 1]

14 Oct 2022

PONE-D-22-16356R1Impact of COVID-19 pandemic on the psychological outcomes: multiple indicators and multiple causes (MIMIC) modelingPLOS ONE

Dear Dr. Rani,

Thank you for submitting your manuscript to PLOS ONE. After careful consideration, we feel that it has merit but does not fully meet PLOS ONE’s publication criteria as it currently stands. Therefore, we invite you to submit a revised version of the manuscript that addresses the points raised during the review process. Please Provide strobe statement as per cross sectional study and prepare your manuscript according to strobe guideline and submit the checklist as per STROBE criteria and attached as supplementary file. Please submit your revised manuscript by 14th November 2022. If you will need more time than this to complete your revisions, please reply to this message or contact the journal office at plosone@plos.org. Please include the following items when submitting your revised manuscript:A rebuttal letter that responds to each point raised by the academic editor and reviewer(s). You should upload this letter as a separate file labeled 'Response to Reviewers'.A marked-up copy of your manuscript that highlights changes made to the original version. You should upload this as a separate file labeled 'Revised Manuscript with Track Changes'.An unmarked version of your revised paper without tracked changes. You should upload this as a separate file labeled 'Manuscript'.If applicable, we recommend that you deposit your laboratory protocols in protocols.io to enhance the reproducibility of your results. Protocols.io assigns your protocol its own identifier (DOI) so that it can be cited independently in the future. For instructions see: https://journals.plos.org/plosone/s/submission-guidelines#loc-laboratory-protocols. Additionally, PLOS ONE offers an option for publishing peer-reviewed Lab Protocol articles, which describe protocols hosted on protocols.io. Read more information on sharing protocols at https://plos.org/protocols?utm_medium=editorial-email&utm_source=authorletters&utm_campaign=protocols.

We look forward to receiving your revised manuscript.

Kind regards,

Muhammad Shahzad Aslam, Ph.D.,M.Phil., Pharm-D

Academic Editor

PLOS ONE

Journal Requirements:

Additional Editor Comments:

Please Provide strobe statement as per cross sectional study and prepare your manuscript according to strobe guideline and submit the checklist as per STROBE criteria and attached as supplementary file

---

## [Author Response · Author response to Decision Letter 1]

22 Oct 2022

Dear Prof Dr. Muhammad Shahzad Aslam,

Academic Editor,

PLOS ONE,

First, we express our extreme gratitude for your extraordinary efforts and incredible handling and reviewing of our revised manuscript after implementing the valuable and constructive Editor's and reviewers' comments and suggestions raised in the first round of the revision to improve the accuracy and level of the manuscript; we greatly appreciate it.

Dear Prof, we have received your decision on the revised manuscript with extreme happiness and respect. Regarding your additional minor comment on the revised manuscript version, we are delighted to submit the revised manuscript R2 entitled “A cross-sectional study on the impact of the COVID-19 pandemic on psychological outcomes: multiple indicators and multiple causes modeling” after implementing your comment to provide strobe statement as per cross-sectional study and to prepare the manuscript according to strobe guideline and submit the STROBE checklist as a supplementary file. Accordingly, we have prepared our revised manuscript according to the strobe statement and submitted the checklist as a supplementary file. Moreover, We have checked and reviewed the reference list and ensured that it is complete and correct. In addition, we have done proofreading for the entire manuscript.

We are grateful to you, and We have implemented the Journal Requirement and Editor's comment with extreme happiness. Updates are highlighted in yellow in the second revised manuscript. Also, here is our Response to the academic editor.

Thank you again for your extraordinary efforts and wonderful and constructive editor of our manuscript. 

We greatly appreciate it.

Journal Requirements:

Please review your reference list to ensure that it is complete and correct

Response: 

Thank you very much for your time and consideration in handling and checking our manuscript; we greatly appreciate it. We have checked and reviewed the reference list and ensured that it is complete and correct. 

Additional Editor Comments:

Please Provide strobe statement as per cross sectional study and prepare your manuscript according to strobe guideline and submit the checklist as per STROBE criteria and attached as supplementary file.

Response: 

Thank you very much for your comment. We have prepared our revised manuscript according to the strobe statement and submitted the checklist as a supplementary file.

Thank you very much once again for your extraordinary efforts and incredible handling and reviewing of our manuscript, 

We highly appreciate it.

Yours sincerely,

---

## [Editor Report · Decision Letter 2]

26 Oct 2022

A cross-sectional study on the impact of the COVID-19 pandemic on psychological outcomes: multiple indicators and multiple causes modeling

PONE-D-22-16356R2

Dear,

We’re pleased to inform you that your manuscript has been judged scientifically suitable for publication and will be formally accepted for publication once it meets all outstanding technical requirements.

Kind regards,

Muhammad Shahzad Aslam, Ph.D.,M.Phil., Pharm-D

Academic Editor

PLOS ONE